# Advanced Glycation End Products and Psoriasis

**DOI:** 10.3390/vaccines11030617

**Published:** 2023-03-08

**Authors:** Martina Maurelli, Paolo Gisondi, Giampiero Girolomoni

**Affiliations:** Section of Dermatology and Venereology, Department of Medicine, University of Verona, 37126 Verona, Italy

**Keywords:** psoriasis, advanced glycation end products, AGEs, inflammation, autoimmunity

## Abstract

Advanced glycation end products (AGEs) are biologically active compounds formed physiologically throughout a sequence of chemical reactions, to generate highly oxidant-reactive aldehydes that combine covalently to proteins. They accumulate slowly in tissues during ageing but also in metabolic and selected inflammatory disorders. Accumulation of AGEs occurs more rapidly and intensely in the skin and serum of patients with type 2 diabetes, obesity, cardiovascular diseases, chronic renal insufficiency, and non-alcoholic fatty liver disease and also in the skin of patients with psoriasis. All of the above conditions are intimately associated with psoriasis. Interaction of AGEs with their receptors (RAGEs) stimulates cellular signaling with the formation of reactive oxygen species and activation of nuclear factor kappa light chain enhancer of activated B (NF-kB), which is a key regulator in the expression of inflammatory mediators and the production of oxidative stress. Thus, AGEs may play an interesting pathogenic role in the intersection of inflammatory and metabolic diseases, may represent a biomarker of inflammation and a potential target for novel therapeutic strategies. This is a narrative review with the objective to summarize current evidence on the role of AGEs in psoriasis.

## 1. Introduction

Psoriasis is a chronic, immune-mediated skin disease that affects 1–4% of the Western countries’ population and about 14 million people in Europe [1,2,3]. Psoriasis is characterized by erythematous–desquamative plaques that typically affect the extensor surfaces such as elbows, knees, sacral region, and scalp. About 20–30% of patients with chronic plaque psoriasis have a moderate to severe disease characterized by extensive skin involvement (i.e., greater than 10% of the body surface area). Both genetic and environmental factors play a relevant role in psoriasis pathogenesis. Most psoriasis susceptibility loci are related to inflammatory and immunity genes [4,5,6]. Environmental and triggering factors include infections (including COVID), stressful life events, medications, skin trauma, and air pollution [4,5,6]. Early events in the development of psoriasis lesions include the recruitment and the activation of plasmacytoid dendritic, which together with natural killer cells and macrophages, secrete cytokines such as IFN-γ, IL-1β, and TNF-α. These cytokines activate myeloid dendritic cells to perform their migratory and antigen-presenting cell functions. Myeloid dendritic cells migrate to lymph nodes to activate naïve autoreactive T lymphocytes, which acquire the ability to recirculate in the skin microenvironment. Activated dendritic cells and macrophages secrete IL-23, a cytokine crucial for the survival and expansion of Th17 helper lymphocytes. These Th17 cells release the pathogenetic cytokines, IL-17, TNF-α and IL-22, which promote directly and indirectly the hyperproliferation of keratinocytes, and the release of chemokines and cytokines. The amplified inflammatory response, and the increased epidermal thickness and desquamation result in the typical skin changes of psoriasis plaque [4,7].

Psoriasis shares immunopathological pathways and genetic features with psoriatic arthritis (PsA), inflammatory bowel diseases (IBD), and uveitis. These diseases are frequently observed in patients with psoriasis. Moreover, psoriasis patients, particularly those with moderate to severe disease, are at greater risk of metabolic disorders, including type 2 diabetes, obesity, chronic renal insufficiency, non-alcoholic fatty liver disease (NAFLD), and the metabolic syndrome. The association between psoriasis and metabolic comorbidities is multifactorial and includes a common genetic background and environmental factors such unhealthy life habits and meta-inflammation [8,9]. Psychiatric diseases (depression, anxiety, suicidal ideation) have also been associated with severe psoriasis [1,2,3,4].

Advanced glycation end products (AGEs) are biologically active oxidant compounds that accumulate in tissues during ageing but also in metabolic and selected inflammatory disorders. AGEs act as pro-inflammatory mediators, and their accumulation occurs more rapidly and intensely in the skin and serum of patients with metabolic disorders and also in the skin of patients with psoriasis. Thus, AGEs may play a pathogenic role in the intersection of inflammatory and metabolic diseases [1,2,3]. In this narrative review, we highlight the role of AGEs in psoriasis.

## 2. Metabolic Comorbidities of Psoriasis

Psoriasis patients are frequently overweight or obese [10]. Obesity, body mass index, hip circumference, and waist–hip ratio are independent risk factors for psoriasis [11]. The risk increases with obesity severity, and there is also a direct correlation between psoriasis severity and obesity. Arterial hypertension is also more prevalent in psoriatic patients compared to the reference population [12]. Poorly controlled and severe hypertension appear to be more common in those patients with more severe psoriasis. Psoriasis is also an independent risk factor for type 2 diabetes mellitus, with a higher risk in patients with more severe skin disease [13]. In addition, diabetic patients with psoriasis are more likely to have micro- and macro-vascular complications, compared to patients without psoriasis. An atherogenic lipid profile, with high total cholesterol and reduced high-density lipoprotein (HDL), is more frequent in patients with psoriasis [14]. In addition, dyslipidemia is a risk factor for developing psoriasis. Metabolic syndrome describes the combination of central obesity, arterial hypertension, insulin resistance, and dyslipidemia. Metabolic syndrome is a very relevant risk factor for diabetes and cardiovascular diseases. The prevalence of metabolic syndrome is higher in patients with moderate to severe psoriasis compared to matched controls, and correlates directly with body surface area affected by psoriasis [15]. Analysis of each component of metabolic syndrome showed the strongest association with obesity, suggesting that the adiposity is the main factor in the association between psoriasis and metabolic syndrome. Although the major cardiovascular risk factors are more prevalent in psoriatic patients, moderate to severe psoriasis is an independent risk factor for major cardiovascular diseases, with the risk of myocardial infarction, stroke, and cardiovascular death greatest among patients with severe psoriasis and longer disease duration. NAFLD is commonly found in the general population and comprises a range from mild forms of steatosis to the more severe steato-hepatitis. Psoriasis is frequently associated with metabolic disorders that can favor liver steatosis. Indeed, patients with psoriasis have a higher prevalence of NAFLD compared with matched non-psoriatic patients, and psoriasis itself is an independent risk factor for NAFLD [16]. Finally, moderate-to-severe psoriasis may be an independent risk factor for chronic kidney disease and end-stage renal disease [9]. 

PsA is the major inflammatory comorbidity associated with psoriasis. The prevalence of PsA is around one third of Caucasian patients with psoriasis, in the range of 6–42%, and is highest among patients between 30 and 60 years [17]. It is possible for patients to present with psoriasis (i.e., skin lesions) without psoriatic arthritis and with psoriatic arthritis without psoriasis skin lesions. The severity of PsA is independent from the severity of psoriasis PsA, which has in most cases mild to moderate disease with a fluctuating course. However, bone erosions can occur early in the disease’s course and lead to disability and impaired function of affected joints and bones [18]. The Group for Research and Assessment of Psoriasis and Psoriatic Arthritis (GRAPPA) has recognized different domains of PsA including synovitis, enthesitis/dactylitis, nail, skin, and axial involvement [19]. Synovitis could potentially affect any joint, but it is more commonly observed in the hands or feet. Enthesitis is observed in 30–50% of patients and more frequently affects the plantar fascia and Achilles’ tendon, causing pain and/or functional impairment, and more rarely in the iliac crest and epicondyles. Dactylitis affects 40–50% of patients, involving either fingers and/or toes asymmetrically [20]. 

Psoriasis can also be associated with IBD, particularly with Crohn’s disease [21]. Patients with psoriasis and IBD are at a higher risk of comorbidities (seronegative arthritis, thyroiditis, diabetes, and lymphoma) than patients affected with only psoriasis. Psoriasis has also been associated to a lesser extent with other diseases, such as cancer, especially T-cell lymphoma, psychiatric disorders, chronic pulmonary disease and obstructive sleep apnea, peptic ulcer disease, hyperuricemia and gout, and osteoporosis [9].

## 3. Advanced Glycation End Products (AGEs)

AGEs are highly oxidant, cross-linked irreversible ketone adducts created by a post-translational non-enzymatic modification between macromolecules, such as free amino groups of proteins, lipids or nucleic acids, and saccharides, such as glucose, fructose, and pentose [1,2,3,22]. Protein glycation is induced by the Maillard reaction in three steps: 1. formation of Schiff bases; 2. early creation of unstable AGE precursors that may trigger Amadori rearrangement; 3. the formation of late irreversible AGE products (Figure 1) [20,23]. Amadori compounds are non-reversible and a stable ketoamines that react with amino groups of proteins. Amadori products subsequently undergo a series of complex rearrangements yielding to AGEs. These rearrangements include oxidative degradation, which generates highly reactive intermediate dicarbonyl compounds, called glyoxal, methyglyoxal (MG), and deoxyglucosones. The principal AGEs are N-carboxylethylysine (CEL), N-carboxymethylysine (CML), N-lactatolysine, pyrraline, and pentosidine, and they have all been investigated in patients with psoriasis as well (Table 1). AGEs are formed primarily from glucose. On the contrary, advanced lipoxidation end products (ALEs) are generated from lipid peroxidation through a non-enzymatic reaction between reactive carbonyl species, the nucleophilic residues of macromolecules. ALEs exert their harmful bioactivity as a result of covalent modification of their target proteins and enzymes, which may result in changes in their biologic functions [2,3,24,25,26,27,28]. AGEs and ALEs are produced from the same precursors (glyoxal and methylglyoxal) and through the same intermediates (CML and N-carboxymethyl-cysteine-CMC). ALEs are generated from lipid peroxidation reactions, and AGEs from glycoxidation reactions [2,3,24,25,26,27,28]. Formation of endogenous AGEs is a physiological process in the normal metabolism over a lifetime and they accumulate slowly in tissues during ageing. In the case of hyperglycemia, obesity, selected autoimmune and inflammatory diseases, chronic renal insufficiency, and enhanced oxidative or carbonyl stress, there is a more rapid and intense accumulation of AGEs [24]. Exogenous AGEs derive from foods, called dietary AGEs, ultraviolet irradiation and ionizing radiation, air pollution, and cigarette smoking. Dietary AGEs are formed from Maillard’s reaction (Figure 1) and may be amplified 10–100 times during the cooking process involving high dry-heat temperatures in grilled and fried food, in particular, in the repeated heating of cooking oil as well as in red meat [24,29]. The oxidative stress deriving from skin exposure to ultraviolet light and smoking habit promotes AGEs accumulation by a greater production of oxygen free radicals [24].

The receptors for AGEs (RAGEs) belong to the immunoglobulin superfamily. They are expressed on the surface of many cell types, including epithelial cells (keratinocytes, hepatocytes), dendritic cells, endothelial cells, and macrophages. RAGEs are also present in the serum in a soluble form [24,30].

## 4. Methods of Assessing AGEs 

Several methods for quantifying AGEs have been developed without any real consensus on a gold standard technique. AGEs can be measured in biological fluids or in tissue samples, in particular, on serum or plasma [31]. The use of urinary or saliva samples has been less common. The evaluation of tissue AGEs poses the main problem of obtaining tissue samples, which may require invasive procedures, such as skin biopsies. The patients are usually more reluctant to give their consent to a skin biopsy than to provide blood samples. To overcome this problem, non-invasive measurement methods have been developed, such as skin autofluorescence (SA) (Table 2) [32,33,34]. It is still difficult to compare results in AGE quantification from different studies because of the different modes of AGE measurements and the lack of method standardization.

### 4.1. Serum Detection with Enzyme-Linked Immunoassay (ELISA)

AGEs can be easily measured in the serum. In particular, serum total AGEs are derived from venous sample collection, centrifuged, and subsequently measured using enzyme-linked immunoassay (ELISA) kit. In addition, the soluble RAGEs can be measured throughout serum using the ELISA technique and comprise both the extracellular domain of wild-type full-length RAGE and the endogenous-secreted isoform lacking a transmembrane domain (esRAGE) [2,3]. As glycation occurs continuously over the lifetime of the protein, the AGEs’ concentration reflects the average blood glucose value over a period of time [32].

### 4.2. Liquid Chromatography Coupled to Tandem Mass Spectrometry (LC-MS/MS)

Liquid chromatography coupled to tandem mass spectrometry (LC-MS/MS) methods are the gold standard, where AGEs are measured throughout serum, and serve as reference methods owing to their excellent analytical performances. Nevertheless, they are not easy to perform in daily practice or in large-scale studies. The results of LC-MS/MS assays are expressed in mmol/mol of lysine. Such unit of measure is used because protein glycation in biological systems occurs on lysine, arginine, and N-terminal residues of proteins [32] and lysine is a substrate of the glycation.

### 4.3. Cutaneous AGEs

Cutaneous AGEs can also be measured as skin autofluorescence (SA) with the AGE ReaderTM. This is a standardized, non-invasive, and reproducible technique, and closely reflects serum AGEs values [1,2,3]. Detection of AGEs with SA is generally performed on the forearm [32]. A limitation of this method is that skin pigments, such as melanin, may interfere with the quantification, which limits its use in individuals with a darker skin complexion [32]. Cutaneous AGEs are predominantly bound to skin collagen, which has been estimated to have a half-life of approximately 14 years; therefore, their levels are assumed to remain quite stable over time. The results of SA values are expressed in arbitrary units [32].

## 5. AGEs and Psoriasis

### 5.1. AGEs in Psoriasis 

AGEs have been investigated in patients with psoriasis and this disease may represent a model to exemplify how AGEs may be associated not only with metabolic but also inflammatory diseases. Papagrigoraki et al. showed that levels of skin AGEs were significantly higher in patients with severe psoriasis (16.84 × 10^3^ μg/mL) compared to patients with mild psoriasis (6.33 × 10^3^ μg/mL), and age, sex, and body mass index matched controls with severe atopic eczema (14.81 × 10^3^ μg/mL) or healthy individuals (12.29 × 10^3^ μg/mL) [2], suggesting a specific accumulation of these compounds in psoriasis (Th17 dominated) as compared to atopic dermatitis (Th2 dominated). No differences in levels of AGEs between normal and lesional skin in psoriasis and eczema patients have been detected. A direct correlation between serum and/or cutaneous AGEs levels and psoriasis severity was also shown. Cutaneous AGEs (*p* < 0.04), serum AGEs (*p* < 0.03), and pentosidine (*p* < 0.05) were higher in patients with severe psoriasis. Cutaneous AGE levels correlated with serum AGEs (r = 0.93, *p* < 0.0001) and with psoriasis area and severity score (r = 0.91, *p* < 0.0001). RAGE levels were lower (*p* < 0.001) in severe psoriasis, and inversely correlated with disease severity [35]. In other studies, serum amount of pentosidine and psoriasin were shown to be significantly higher in patients with severe psoriasis compared to patients with mild psoriasis or eczema, and healthy individuals. In these studies, controls and patients with severe AD were also matched for age, sex, and BMI in order to minimize the selection bias [36,37]. The amount of both compounds correlated with psoriasis severity [2]. Psoriasin is a proinflammatory protein that belongs to the S100 family of small calcium-binding proteins. Psoriasin can bind RAGEs and stimulates further production of AGEs. A direct correlation between intima-media thickness and serum psoriasin levels has been shown in patients with psoriasis [36]. Furthermore, a direct correlation between psoriasis severity and serum levels of koebnerisin has been described [38]. Instead, serum levels of RAGEs were lower in patients with psoriasis compared to patients with eczema or healthy subjects. RAGEs levels correlated inversely with disease severity [2]. More recently, we found that effective treatment of moderate to severe psoriasis with biological agents is associated with significant reduction in the levels of cutaneous AGEs [1]. In particular, a marked and sustained improvement in psoriasis with IL-17 and IL-23 inhibitors was associated with marked decrease in cutaneous AGEs. To what extent this effect may also have a favorable impact on metabolic comorbidities needs further investigation. Our findings are consistent with previous studies that have shown a reduction in SA in patients with psoriasis or PsA treated with adalimumab [39], or in patients with inactive disease [40]. Damasiewicz-Bodzek A et al. showed that blood concentration of AGEs is higher during the active disease phase and decreases during remission [40]. 

In all these studies, those patients with a smoking habit, diabetes, dyslipidemia, hypercholesterolemia, hypertension and/or any other systemic inflammatory or autoimmune diseases were excluded to avoid confounding factors affecting levels of AGEs [36,37,38,39]. Having methodologically excluded these confounding factors, it could be speculated that the association between AGEs and psoriasis is linked to the specific inflammation of psoriasis. AGEs act both directly on skin tissue and through their RAGE receptors. Accumulation of AGEs in the skin may cause an increased production of free radicals, oxidized LDL, and consequently favor skin peroxidation. Proteins modified by glycation or glycosylation can become immunogenic, thus amplifying pathogenic immune responses [41]. The AGE–RAGE interaction on inflammatory cells such as monocytes, macrophages, neutrophils, and endothelial cells stabilizes the receptor in the active state and favors the release of cytokines and chemokines, the production of reactive oxygen species, and the activation of metalloproteases and migration of T lymphocytes and other inflammatory cells into the inflammation site [41,42,43]. RAGEs can also bind the S100 proteins, including psoriasin and koebnerisin [2,3]. To what extent these phenomena may play a role in the pathogenetic mechanisms of psoriasis needs to be investigated. 

### 5.2. AGEs as Biomarker of Cardiovascular Risk in Patients with Psoriasis

Moderate to severe psoriasis has been associated with an increased cardiovascular risk. In particular, an increased relative risk of myocardial infarction (1.17; 95% CI 1.11–1.24) and cardiovascular death (1.46; 95% CI 1.26–1.69) has been reported in patients with severe psoriasis [44]. This may be related to the association with classical cardiovascular risk factors including type 2 diabetes, obesity, hypertension, chronic renal insufficiency, and NAFLD (Table 3). In all these disorders, higher levels of AGEs have been measured [44]. The interaction of AGEs with RAGE in endothelial and smooth muscle cells as well as in platelets, activates intracellular signaling, and leads to endothelial cell injury, modulation of vascular smooth muscle cell function, and altered platelet activity [44]. In addition, the activation of RAGE can induce complex signaling pathways leading to enhanced calcium deposition, and increased vascular smooth muscle apoptosis, concurring to the development of atherosclerosis [45]. A relationship between psoriasis, AGEs, and atherosclerosis has been studied. Ergun T et al. found that SA was significantly increased in psoriasis patients (N = 52) compared to healthy controls (N = 20) and correlated with carotid intima-media thickness, supporting the hypothesis of a role for AGEs in atherosclerosis in patients with psoriasis [46]. Batycka-Baran A et al. found decreased soluble RAGE levels in patients with chronic plaque psoriasis compared to controls and an inverse correlation with psoriasis severity as assessed with the psoriasis area severity index. This correlation was independent of other variables. They concluded that decreased levels of soluble RAGE may contribute to the chronic inflammatory process and atherosclerosis [47]. Measuring cutaneous AGEs by SA may be a surrogate marker of systemic AGE burden as well as systemic inflammation and may thus represent a non-invasive method to identify patients with psoriasis with an increased cardiovascular risk [48].

## 6. AGEs in Metabolic Diseases

Comorbidities associated with psoriasis, including metabolic disorders such as type 2 diabetes, obesity, hypertension, chronic renal insufficiency, and NAFLD, have been linked to increased serum levels of AGEs. An increased AGEs formation and accumulation in human tissues may cause modifications in the structure and function of the cells, leading to the amplification of inflammation, accelerating the development of the comorbidities associated with psoriasis, and correlating with their prognosis [2,3,24,58]. In particular, obesity and hyperglycemia strongly contribute to AGE production and tissue accumulation [8,36]. Indeed, serum levels of AGEs are predictors of heart failure and the development of cardiovascular events. AGEs increase vascular permeability, and arterial stiffness and promote neoangiogenesis. In addition, they inhibit vasodilation by interfering with nitric oxide, and thus promote endothelial cell dysfunction, ultimately favoring atherosclerosis. In patients with type 2 diabetes, serum levels of AGEs correlate with the risk and severity of cardiovascular disease. Higher expression of RAGE is associated with a stronger inflammation in the atherosclerotic carotid plaque, and treatment with statins decreases inflammation and expression of RAGE [59]. The activation of the AGE–RAGE axis causes the liberation and accumulation of reactive oxygen species (ROS) and activation of nuclear factor kappa light chain enhancer of activated (NF-κB), which is a master regulator of inflammation [59]. The binding of AGEs to RAGEs also modifies LDLs and promotes their uptake by macrophages and thus foam cell formation. AGEs are involved in the pathomechanism of diabetes, and they may be a crucial diagnostic marker, due to the long time remaining in the body. Their determination may allow for monitoring the progression of the disease and the effectiveness of the treatment [60]. In addition, measuring cutaneous AGEs in diabetic patients has been proposed to be important to identify those who have a major risk in developing vascular complications. High SA is an independent marker of acute myocardial infarction, cardiovascular disease, inflammation, and oxidative stress; it is also a strong predictor of survival in diabetic patients [2,3,24,58].

## 7. Conclusions

AGEs are biologically active products involved in the amplification and perpetuation of chronic immune-mediated inflammatory process possibly by the sustained activation of NF-κB. RAGE-mediated signaling triggers the expression of several inflammatory mediators, including adhesion molecules, chemokines, proinflammatory cytokines, such as TNF-α, IL-1, -6, and -8, and the consequent increased recruitment of inflammatory cells to the site of inflammation [2,3]. AGE formation and accretion in human tissues may cause modifications with a prominent role in the pathophysiology of inflammatory and metabolic diseases. In patients with psoriasis, the intensified protein glycation in the skin may have a role in the amplification of skin inflammation and may provide a link between cutaneous inflammation and increased prevalence of associated metabolic diseases. Increasing evidence supports a possible role for the AGE/RAGE axis also in the development, severity, and progression of cardiovascular diseases in patients with psoriasis. Additional prospective multicenter randomized controlled studies are needed to further evaluate the possibility that circulating AGEs or soluble RAGE levels can serve as a biomarker for cardiovascular diseases risk and to identify patients who may benefit from early prevention and treatment. More convincingly, AGEs and RAGEs may become a biomarker of systemic inflammation, metabolic abnormalities, and cardiovascular risk in patients with psoriasis. Moreover, AGEs and RAGEs may represent a potential therapeutic target, and investigating the role of AGEs in the crosstalk between psoriasis and its metabolic comorbidities may pave the way to new therapeutic targets and strategies.

## Figures and Tables

**Figure 1 vaccines-11-00617-f001:**
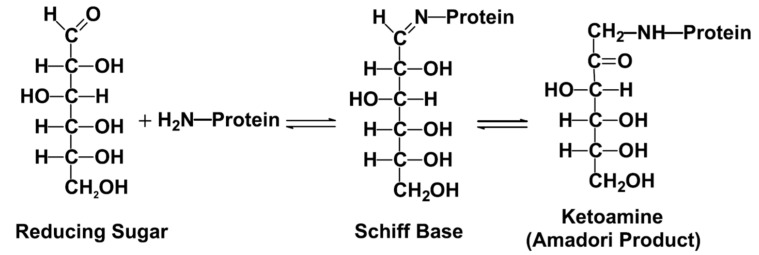
The classical pathway of the Maillard reaction. The open chain form of glucose reacts with lysine to form a Schiff base adduct, which undergoes an Amadori rearrangement to form a ketoamine (fructosamine) adduct to protein [23].

**Table 1 vaccines-11-00617-t001:** Principal advanced glycation end products (AGEs) investigated in patients with psoriasis.

AGEs	Other RAGE Ligands
Carboxymethylysine (CML)	Psoriasin (S100A7)
Carboxyethyllysine (CEL)	Koebnerisin (S100A15)
Pyrraline	High-mobility group protein B1 (HMGB1)
Pentosidine	
Glyoxal-lysine dimer (GOLD)	
Deoxyglucasone-lysine dimer (DOLD)	
Methyl glyoxal-lysine dimer (MOLD)	
N-lactatolysine	

**Table 2 vaccines-11-00617-t002:** Characteristics of the main analytical methods currently used for AGEs quantification.

Methods	AGEs	Sample
ELISA	Carboxymethylysine (CML)	Blood
	Total AGEs	Blood
LC-MS/MS	Carboxymethylysine (CML)	Blood
	AGEs panels	Blood
SA	Total fluorescent AGEs	Skin

AGEs, advanced glycation end products; LC-MS/MS, liquid chromatography coupled to tandem mass spectrometry; ELISA, enzyme-linked immunoassay; SA, skin autofluorescence.

**Table 3 vaccines-11-00617-t003:** Psoriasis comorbidities associated with an increased cardiovascular risk.

Comorbidities	Measure of Association * between Psoriasis and Comorbidities
Obesity	20.7% (OR, 1.79; 95% CI, 1.55–2.05) [49,50,51]
Type 2 diabetes	7.0–11.4% [52]
Metabolic syndrome	20–50% (OR, 2.26; 95% CI, 1.70–3.01) [53]
Dyslipidemia	62.8% (OR, 1.04–5.55; 95% CI, 3.53–10.36) [54]
Non-alcoholic fatty liver disease	47% (OR, 1.7; 95% CI, 1.1–2.6) [53]
Chronic renal insufficiency	15.2% [55,56,57]

* Prevalence and/or odds ratio of the association.

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
