# Peer review of "Advanced Glycation End Products and Psoriasis"

_vaccines, 2023, doi:10.3390/vaccines11030617_

Round 1

Reviewer 1 Report

This Review on Advanced Glycation End products (AGEs) and Psoriasis and comorbidities is well written and ready to be published.

Author Response

Thank you very much for Your response.

Reviewer 2 Report

The manuscript entitled ‘Advanced glycation end products and psoriasis’ is a review on the role of AGEs in psoriasis.

The purpose of the manuscript is to report the relationship between AGEs and psoriasis, taking into account also cardiovascular diseases.

Point of discussion

- L.64-64 The sentence is not useful and can be erased.

 Section 3 - Advanced glycation end-products

 L.115-126 The description of AGE formation is not easy to follow; could the authors add the chemical structures of the compounds and the reactions of the steps 1, 2 and 3? The sequence of chemical reactions must be added.

L.130-132. What are the differences between AGEs and ALEs? This point should be improved.

 Section 4 - Methods of assessing AGEs

The sentence “Although biological fluids are easy to obtain, they provide only an instant view of AGE accumulation, because of the short average half-life of proteins.” Is not clear; it should be improved.

Also, some references have to be added to support the introduction of this section (L.153-158).

“The results of ELISA assay are usually in ng/mL”; could the authors report the range of the AGEs in the serum from health subjects and patients?

L.173-175 “The results of LC-MS/MS assays are expressed in mmol/mol of lysine”; What is the meaning? Perhaps reporting some examples could be useful. Similar considerations for sentence reported in L. 183.

L.251 The acronym “RASI” should be explained.

L.277-278 The sentence needs of some references to support it.

Author Response

2215160 - Advanced glycation end products and psoriasis

Point by point response to reviewer #2

REVIEWER COMMENTS

The manuscript entitled ‘Advanced glycation end products and psoriasis’ is a review on the role of AGEs in psoriasis. The purpose of the manuscript is to report the relationship between AGEs and psoriasis, taking into account also cardiovascular diseases.

L.64-64 The sentence is not useful and can be erased.

AUTHORS REPLY

The sentence L.64-64 has been erased, as requested by the Reviewer.

REVIEWER COMMENTS

Section 3 - Advanced glycation end-products. L.115-126 The description of AGE formation is not easy to follow; could the authors add the chemical structures of the compounds and the reactions of the steps 1, 2 and 3? The sequence of chemical reactions must be added.

AUTHORS REPLY

The chemical structures of the compounds and the reactions of the steps 1, 2 and 3 have been included, as requested by the Reviewer.

REVIEWER COMMENTS

L.130-132. What are the differences between AGEs and ALEs? This point should be improved.

AUTHORS REPLY

The differences between AGEs and ALEs have been described (line 137-140) and references included. In particular, “AGEs and ALEs are produced from the same precursors (glyoxal and methylglyoxal) and through the same intermediates (CML and N-carboxymethyl-cysteine-CMC). ALEs are generated from lipid peroxidation reactions, and AGEs from glycoxidation reactions [2,3,24,25-28].”

REVIEWER COMMENTS

Section 4 - Methods of assessing AGEs. The sentence “Although biological fluids are easy to obtain, they provide only an instant view of AGE accumulation, because of the short average half-life of proteins.” Is not clear; it should be improved.

Also, some references have to be added to support the introduction of this section (L.153-158).

AUTHORS REPLY

The sentence has been rephrased in order to make it clearer. Two new references have been included.

  1. Perrone, A.; Giovino, A.; Benny, J.; Martinelli, F. Advanced Glycation End Products (AGEs): biochemistry, signaling, analytical methods, and epigenetic effects. Oxid Med Cell Longev 2020, 2020:3818196.
  2. Jaisson, S.; Gillery, P. Methods to assess advanced glycation end-products. Curr Opin Clin Nutr Metab Care 2021, 24:411-5.

REVIEWER COMMENTS

“The results of ELISA assay are usually in ng/mL”; could the authors report the range of the AGEs in the serum from health subjects and patients?

AUTHORS REPLY

The value of the AGEs in the serum in healthy subjects, in patients with psoriasis and in patients affected by eczema have been reported (line 225-226).

REVIEWER COMMENTS

L.173-175 “The results of LC-MS/MS assays are expressed in mmol/mol of lysine”; What is the meaning? Perhaps reporting some examples could be useful. Similar considerations for sentence reported in L. 183.

AUTHORS REPLY

Such unit of measure is used because protein glycation in biological systems occurs on lysine, arginine and N-terminal residues of proteins [32] and lysine is a substrate of the glycation.

REVIEWER COMMENTS

L.251 The acronym “RASI” should be explained.

AUTHORS REPLY

The sentence has been rephrased as follows: “A direct correlation between serum and/or cutaneous AGEs levels and psoriasis severity has also been shown”.

REVIEWER COMMENTS

L.277-278 The sentence needs of some references to support it.

AUTHORS REPLY

The references [36-39] have been included and cited. 

Reviewer 3 Report

In this review Maurelli et al. examine data in the literature concerning advanced glycation endproducts (AGEs) and psoriasis. AGEs accumulate in tissues with aging, and with certain diseases, such as diabetes, obesity and other inflammatory conditions, this process can be accelerated. Thus, it seems possible that there is some involvement of AGEs in the phenotypic expression of psoriasis. However, as these authors note, patients with psoriasis often exhibit diseases such as diabetes and obesity associated with higher AGEs, making it more difficult to distinguish between effects of psoriasis itself and these other comorbidities. The review is timely and correctly emphasizes the need for additional studies, but there are some opportunities for improvement as delineated below.

Minor points:

(1) Section 2 on metabolic comorbidities of psoriasis is not well referenced, with most assertions made without citations to strengthen them. The authors should provide additional citations for statements in this section.

(2) It is possible for patients to present with psoriasis (i.e., skin lesions) without psoriatic arthritis and with psoriatic arthritis without psoriasis skin lesions. Therefore, some of the sentences in the second paragraph of section 2 do not seem quite accurate and should be revised. 

(3) The sentence in lines 193-194 seems opposite to the sentence before (lines 189-192), and this disparity must be addressed.

(4) The sentence in lines 230-231 seems redundant with the statements in lines 200-201.

(5) In the tables and text, lysine is spelled with a y.

(6) In the final line, the saying is “pave the way”.

(7) All abbreviations should be defined on first use. NF-kappaB is actually “nuclear factor kappa light chain enhancer of activated B cells”.

Author Response

2215160 - Advanced glycation end products and psoriasis

Point by point response to reviewer #3

REVIEWER COMMENTS

In this review Maurelli et al. examine data in the literature concerning advanced glycation end products (AGEs) and psoriasis. AGEs accumulate in tissues with aging, and with certain diseases, such as diabetes, obesity and other inflammatory conditions, this process can be accelerated. Thus, it seems possible that there is some involvement of AGEs in the phenotypic expression of psoriasis. However, as these authors note, patients with psoriasis often exhibit diseases such as diabetes and obesity associated with higher AGEs, making it more difficult to distinguish between effects of psoriasis itself and these other comorbidities. The review is timely and correctly emphasizes the need for additional studies, but there are some opportunities for improvement as delineated below.

Minor points:

  • Section 2 on metabolic comorbidities of psoriasis is not well referenced, with most assertions made without citations to strengthen them. The authors should provide additional citations for statements in this section.

AUTHORS REPLY

A series of references has been included to the section 2 on metabolic comorbidities as kindly suggested by the Reviewer.

REVIEWER COMMENTS

It is possible for patients to present with psoriasis (i.e., skin lesions) without psoriatic arthritis and with psoriatic arthritis without psoriasis skin lesions. Therefore, some of the sentences in the second paragraph of section 2 do not seem quite accurate and should be revised.

AUTHORS REPLY

We fully agree with the comment of the Reviewer and the sentence has been rephrased according to the suggestion.

REVIEWER COMMENTS

The sentence in lines 193-194 seems opposite to the sentence before (lines 189-192), and this disparity must be addressed.

AUTHORS REPLY

The sentences in lines 189-193 have been rephrased in order to address the disparity. Thank for the suggestion.

REVIEWER COMMENTS

The sentence in lines 230-231 seems redundant with the statements in lines 200-201.

AUTHORS REPLY

The sentence in lines 230-231 has been deleted because of redundancy.

REVIEWER COMMENTS

In the tables and text, lysine is spelled with a y.

AUTHORS REPLY

This error has been corrected.

REVIEWER COMMENTS

In the final line, the saying is “pave the way”.

AUTHORS REPLY

The final line has been corrected accordingly.

REVIEWER COMMENTS

All abbreviations should be defined on first use. NF-kappaB is actually “nuclear factor kappa light chain enhancer of activated B cells”.

AUTHORS REPLY

The abbreviation has been corrected accordingly.
